# OpenReview forum: "Conditional Clifford-Steerable CNNs for PDE Modeling"
_ICML.cc/2026/Conference — ICML 2026 regular_

### Official Review · Reviewer_FfRh · 2026-03-09

**Soundness:** 3
**Presentation:** 3
**Significance:** 3
**Originality:** 3
**Overall Recommendation:** 4
**Confidence:** 2

**Summary:**

This paper proposes Conditional Clifford-steerable CNNs (C-CSCNNs), aimed at addressing the limitations in the expressive power of the original CSCNNs. Authors introduce a conditional convolution mechanism, enabling the kernel function to depend not only on spatial offsets but also on the input feature field itself. By incorporating an equivariant pooling operator (e.g., average pooling), the input field is compressed into global features, which serve as additional inputs to the kernel network, allowing it to generate a richer set of kernel bases. Theoretical analysis shows that this approach can recover the missing frequency components under O(2,0) and suggests that its kernel basis is complete.

The experimental section validates the effectiveness of C-CSCNN on several PDE prediction tasks. The results demonstrate that C-CSCNNs significantly outperform standard CSCNNs in terms of data efficiency and prediction accuracy, and perform comparably or better than SOTA methods (such as Transolver) across multiple tasks.

**Compliance With Llm Reviewing Policy:**

Affirmed.

**Final Justification:**

Thank the authors for their response, which addressed my concerns. I maintain my positive recommendation.

**Key Questions For Authors:**

1. For higher-dimensional group structures, such as O(1,3), might the issue of incomplete kernel basis arise?

2. Can the conditioning mechanism be extended to other equivariant networks beyond CSCNNs?

3. Does the conditional convolution mechanism introduce additional computational overhead?

**Limitations:**

yes

**Strengths And Weaknesses:**

**Strenghts**

1. The approach is both elegant and effective: To my knowledge, the conditional convolution mechanism is novel and addresses the limitations in the expressive power of traditional CSCNN. Moreover, the theoretical derivation is rigorous and maintains complete equivariance.

2. The experimental design is comprehensive: The benchmark covers various application scenarios and different spatial dimensions. The authors candidly discuss both the strengths (fluid dynamics) and limitations (electrodynamics) of C-CSCNN.

3. Strong reproducibility: The Appendix provides detailed implementation specifics and code examples.

**Weaknesses**

1. There is no quantitative analysis of the additional overhead introduced by the conditional convolution mechanism.

2. "The kernel basis of conditional Clifford-steerable CNNs is complete" is presented as a conjecture, only illustrated in Appendix A.5 with the case of O(2,0), without a rigorous proof.

---

> ### Author Rebuttal · Authors · 2026-03-31
>
> We thank the reviewer for the positive assessment. We address the weaknesses and questions below.
>
> **1. Incomplete kernel basis for higher-dimensional groups such as O(1,3)**:
>
> Yes, the incompleteness arises for any $\operatorname{O}(p,q)$ with $p+q \geq 2$. Since the initial submission, we have formalised this into a general result that we will include in the camera-ready version. We summarise it here.
> Consider the standard multivector embedding $\iota(x) = \varphi(|x|_\eta) + x$ of a position vector $x \in \mathbb{R}^{p,q}$. The self-product $\iota(x)\,\iota(x)$ decomposes as:
>
> - grade 0: $\varphi^2 + \eta(x,x)$, a function of the pseudo-norm only,
>
> - grade 1: $2\varphi x$, proportional to the input,
>
> - grades $\geq 2$: identically zero.
>
> We prove that this structure persists under arbitrary compositions of geometric self-products and grade-wise linear maps, i.e. all operations available to the kernel network $\mathcal{K}$. Therefore, for any depth and any signature $(p,q)$, the kernel network output satisfies $\mathcal{K}(x)^{(0)} = h_0(|x|_\eta)$ and $\mathcal{K}(x)^{(k)} = 0$ for $k \geq 2$.
>
> Since the vector-vector kernel $K^1{\ 1}(x) = \sum_m \Lambda^1_{m1} w^1_{m1} \mathcal{K}(x)^{(m)}$ depends on precisely these grades (with $\Lambda^1_{11} = 0$ by the selection rules of the geometric product), it collapses to $h_0(|x|_\eta)$ — a scalar function of the norm with no directional dependence. This holds for $\operatorname{O}(1,3)$ just as it does for $\operatorname{O}(2,0)$.
>
> For the conditional kernel network $\hat{\mathcal{K}}$, introducing an auxiliary multivector $\zeta \in \operatorname{Cl}(\mathbb{R}^{p,q})$ with $\langle\zeta\rangle_1 \not\propto x$ yields: grade 0 now contains $\eta(x, \langle\zeta\rangle_1)$, linear in the directional components of $x$, and grade 2 now contains $x \wedge \langle\zeta\rangle_1 \neq 0$. Both channels that were blocked in the standard $\mathcal{K}$ now carry directional information, so $K^1_{\ 1}(x)$ can represent direction-dependent maps. Furthermore, with $L$ independent auxiliary multivectors, degree-$L$ polynomials in the components of $x$ become accessible, enabling recovery of angular frequency $L$.
>
> Note that this result specifically addresses the vector-vector interaction ($k = n = 1$). For higher-dimensional algebras, additional grade-to-grade interactions exist, and proving completeness across all of them for general $\operatorname{O}(p,q)$ remains an open part of the conjecture, as we discussed in the Limitations section. However, the conditioning mechanism enriches all grades simultaneously, and our empirical results on $\operatorname{O}(3,0)$ and $\operatorname{O}(1,2)$ (Table 1) validate that the method works in practice for these higher-dimensional groups.
>
> **2. Extension to other architectures**:
>
> Yes, absolutely. The theoretical framework we outline in Section 4 is generally applicable to any implicitly parameterized steerable kernel. The definitions we provide for the general conditional convolution integral (Definition 4.1), the steerability constraint required to maintain G-equivariance (Lemma 4.2), and the required equivariance of the conditioning operator $T$ (Proposition 4.5) are all agnostic to the choice of the group $G$, the algebra, or the specific network used to parameterize the kernel. As long as the implicit kernel network takes the extracted condition as an auxiliary input and preserves equivariance with respect to the relevant group, this approach can be used to augment and improve the expressivity of any implicit steerable CNN.
>
> **3. Computational overhead**:
>
> We agree that this is a valuable addition to the paper and thus will add it to the camera ready version. The conditional convolution mechanism introduces virtually no computational overhead. We measured inference times on the Maxwell 3D task (single A100 GPU, averaged over the test set):
> | Model | Time (ms/sample) |
> | :--- | :---: |
> | CSCNN | 53.83 |
> | C-CSCNN (Ours) | 54.30 |
> As shown above, C-CSCNN is less than 1% slower than the unconditioned CSCNN. This is because our conditioning mechanism only requires computing a single global spatial mean (mean pooling) of the input field, which is then concatenated to the kernel input. This operation is computationally trivial and adds negligible cost.

---

> > ### Author Rebuttal · Reviewer_FfRh · 2026-04-01
> >
> > Thank the authors for their response, which addressed my concerns. I maintain my positive recommendation.

---

> > > ### Author Response · Authors · 2026-04-02
> > >
> > > Dear reviewer FfRh,
> > >
> > > Thank you for confirming that your concerns have been fully resolved. We are glad that the additional theoretical details and computational overhead metrics addressed your questions, and we will ensure they are highlighted in the camera-ready version. We appreciate your time and constructive feedback throughout the discussion period.
> > >
> > > Sincerely yours,
> > > Authors

---

### Official Review · Reviewer_a8X1 · 2026-03-09

**Soundness:** 3
**Presentation:** 3
**Significance:** 2
**Originality:** 2
**Overall Recommendation:** 5
**Confidence:** 2

**Summary:**

This paper extends Clifford-steerable CNNS by making the convolutional kernel adaptive. Using Clifford algebra, the network can be forced to respect, e.g., rotations of the vector-valued fields such as velocity. However, the authors show that the standard Clifford-steerable CNN has limited representational power. In response, the authors make the kernel adaptive to the feature mean of the whole input field. The experiments show that this tweak can increase the accuracy over the old Clifford-steerable CNN and standard CNNs (ResNet). The model is on par with newer Transformer-based architectures (Transsolver).  The experiments include standard fluid-dynamics tasks as well as Maxwell's equations.

**Compliance With Llm Reviewing Policy:**

Affirmed.

**Final Justification:**

I decided to increase my score. The authors responded well to my initial concerns.  While the proposed change to the Clifford-steerable CNNs is small, the experiments show a strong effect and the paper provides supporting theory.

**Key Questions For Authors:**

1. How does the inference time compare to the baselines (especially the non-equivariant ResNet and the Transsolver)?

**Limitations:**

yes

**Strengths And Weaknesses:**

Strengths
1. The proposed method is a simple but effective tweak to the Clifford-steerable CNN.
2. Evaluation compares to a number of strong baselines.
3. Good presentation and generally well-written.

Weaknesses
1. Rather limited novelty, as the proposed method is a modification of the original Clifford-steerable CNN.
2. The experiments are mostly limited to 1-step evaluation. Looking at longer rollouts would have provided additional insights.

---

> ### Author Rebuttal · Authors · 2026-03-31
>
> We thank the reviewer for acknowledging the effectiveness and strong baselines of our approach. We address the weaknesses and question below.
>
> **Novelty**:
> We note that while the implementation change is indeed lightweight (which we view as a strength), the theoretical contribution involves:
>
> (1) deriving the general equivariance constraint that conditional kernels must satisfy for any group $G$ (Lemma 4.2),
>
> (2) proving that our Clifford-steerable conditional kernels satisfy this constraint (Lemma 4.3),
>
> (3) proving the equivariance conditions on the conditioning operator $T$ required for efficient template matching (Proposition 4.5), and
>
> (4) analyzing how conditioning restores the completeness of the kernel basis (Section 4.4).
>
> We believe that a lightweight, but well-performing implementation backed by highly non-trivial theory is a desirable outcome.
>
> **Long rollout evaluation**:
> We note that we do in fact evaluate multi-step rollouts: the SWE-5 experiment (Table 2, Figure 3) evaluates 5-step autoregressive predictions. We agree that even longer rollouts would provide further insight and are happy to explore this in future work.
>
> **Inference time**:
> We report inference times on the Maxwell 3D task (single A100 GPU, averaged over the test set), which we will add to the camera-ready version:
>
> | Model | Time (ms) |
> | :--- | :---: |
> | Simple ResNet | 1.80 |
> | Transolver | 39.06 |
> | CSCNN | 53.83 |
> | C-CSCNN (Ours) | 54.30 |
>
> Two observations are worth highlighting. First, the mean conditioning mechanism adds virtually no overhead: C-CSCNN is less than 1% slower than the unconditioned CSCNN. Second, our current implementation of Clifford algebra operations (geometric products, multivector processing) is not optimised for speed, but hardware-efficient implementations already exist (e.g., Flash Clifford [7]) and can be integrated to significantly reduce this gap.
>
> [7] Zhdanov, M. (2025). Flash Clifford: Hardware-Efficient Implementation of Clifford Algebra Neural Networks. https://github.com/maxxxzdn/flash-clifford

---

> > ### Author Rebuttal · Reviewer_a8X1 · 2026-04-02
> >
> > Thank you for your response! My concerns have been adequately addressed.

---

> > > ### Author Response · Authors · 2026-04-02
> > >
> > > Dear reviewer a8X1,
> > >
> > > We are sincerely grateful for taking the time to read our rebuttal and for confirming that all concerns have been resolved. We greatly appreciate your support for our work!
> > >
> > > Sincerely yours,
> > > Authors

---

### Official Review · Reviewer_oRKz · 2026-03-13

**Soundness:** 3
**Presentation:** 3
**Significance:** 3
**Originality:** 3
**Overall Recommendation:** 4
**Confidence:** 4

**Summary:**

The paper proposes C-CSCNNs for pde forecasting. The main idea is to improve the expressivity of prior CSCNNs by conditioning their kernels on auxiliary features computed from the input field. The authors derive the corresponding equivariance constraint for the conditional kernels, show how to realize this efficiently, and argue that this addresses the missing kernel basis components in the original model. Then they validate on multiple PDE benchmarks, showing clear gains over standard CSCNNs and competitive performance relative to other strong operator-learning and equivariant baselines.

**Compliance With Llm Reviewing Policy:**

Affirmed.

**Final Justification:**

I would like to maintain my final recommendation of 4 (weak accept). The paper presents a useful idea of using conditional kernels to improve CSCNN expressivity with a solid intuition and consistent empirical gains over the base model.
However my main theoretical concern remains only partially addressed. While their rebuttal strengthens the analysis for the vector-vector interaction, completeness across grade-to-grade interactions is still unresolved. Consequently the broader claim that the method resolves kernel incompleteness remains somewhat overstated. The rebuttal clarifies the scope but does't fully answer this issue or change my overall assessment. I therefore keep my original score, balancing the this remaining gap and the paper's strength.

**Key Questions For Authors:**

1) Can the authors give a convincing analysis for the general O(p, q) case? They provide a convincing analysis for O(2, 0) case but the general case remains open. A strong clarification here would increase my confidence in the theoretical claims of the paper.

2) Can the authors shed some light on the mechanism behind the gains? i am a bit unsure if its because of the mean pooling or because of the recovery of the missing steerable kernel components.

3) The paper notes that parameter counts are matched in all experiments except SWE-5, and it also observes that C-CSCNN trails Transolver on the Maxwell tasks. Can the authors clarify how they expect the method to behave under stricter parameter-matched SWE-5 comparisons, and whether they can comment on why does Transolver behave better than C-CSCNN?

**Limitations:**

Yes.

**Strengths And Weaknesses:**

**Soundness**
The paper is technically well motivated: it identifies a concrete expressivity gap in standard CSCNNs by showing that, in the $O(2, 0)$ case, the vector-to-vector kernel misses the frequency-2 angular component of the analytical solution. The proposed conditional construction is then supported by meaningful theory: the paper derives the equivariance condition for conditional kernels and proves equivariance of both the conditional kernel construction and its efficient pooled implementation. Empirically, the method is evaluated on relevant PDE forecasting benchmarks against strong baselines, includes an equivariance-error check, and shows consistent gains over the original CSCNN.

The shortcoming of the paper is that the strongest claim of the paper is not fully proved in general: beyond the analyzed $O(2, 0)$ setting, kernel-basis completeness is left as a conjecture and supported mainly empirically. Also, the experiments do not fully isolate whether the gains come specifically from recovering missing steerable components or more generally from adding global context through mean-pooling-based conditioning. Finally, SWE-5 is not parameter-matched, making that part of the empirical comparison less clean.

**Presentation**
The submission is generally well written and logically structured: the paper clearly motivates the problem, introduces the limitation of standard CSCNNs, presents the conditional construction, and then evaluates it empirically, so the high-level narrative is easy to follow. It also does a reasonable job positioning itself relative to prior work by discussing equivariant CNNs, Clifford-based networks, implicit kernel parameterizations, and modern PDE-learning baselines, which makes the paper’s intended contribution fairly clear. That said, the presentation is not fully polished at the highest level. The most technical sections, especially the discussion of kernel incompleteness and the conditional equivariance constructions, are fairly dense, and the paper could do a better job distinguishing what is formally proved from what remains conjectural in the main text. I would also have liked a slightly more intuitive explanation of what “missing kernel basis elements” means and a sharper comparison to simpler alternatives, so that readers less familiar with steerable kernels and Clifford algebra can more easily understand why the conditional extension is needed.

**Significance**
The paper addresses a meaningful and relevant problem in equivariant scientific machine learning: how to improve the expressivity of Clifford-Steerable CNNs for PDE forecasting without sacrificing symmetry preservation. That is a worthwhile contribution because CSCNNs are designed for physically structured data on Euclidean and pseudo-Euclidean domains, and the paper identifies a concrete limitation in the prior formulation, kernel incompleteness, that can materially affect performance. The proposed conditional extension appears to improve capabilities in that setting, and the empirical results suggest practical value: the method consistently outperforms standard CSCNNs and is competitive with strong modern baselines on multiple PDE forecasting tasks. I therefore think the work is significant within its niche and likely to influence follow-up research on more expressive equivariant operators for scientific ML. At the same time, the scope of impact is still somewhat specialized rather than broad across all of machine learning, and the paper’s strongest theoretical narrative is not fully closed beyond the analyzed setting, so I would view this as a strong field-specific contribution rather than a paper with very broad significance.

**Originality**
I think the paper addresses a meaningfully original problem within the realm of symmetry enforced PDE forecasting.

---

> ### Author Rebuttal · Authors · 2026-03-31
>
> We thank the reviewer for the thorough and constructive evaluation, and for recognizing our work. We address each question below.
>
>
> **1. General O(p,q) case**:
>
> Since the initial submission, we have formalised the missing kernels and their recovery into a more general result that we will include in the camera-ready version. We summarise it here.
> Consider the standard multivector embedding $\iota(x) = \varphi(|x|_\eta) + x$ of a position vector $x \in \mathbb{R}^{p,q}$. The self-product $\iota(x)\,\iota(x)$ decomposes as:
>
> - grade 0: $\varphi^2 + \eta(x,x)$, a function of the pseudo-norm only,
>
> - grade 1: $2\varphi x$, proportional to the input,
>
> - grades $\geq 2$: identically zero.
>
> We prove that this structure persists under arbitrary compositions of geometric self-products and grade-wise linear maps, i.e. all operations available to the kernel network $\mathcal{K}$. Therefore, for any depth and any signature $(p,q)$, the kernel network output satisfies $\mathcal{K}(x)^{(0)} = h_0(|x|_\eta)$ and $\mathcal{K}(x)^{(k)} = 0$ for $k \geq 2$.
>
> Since the vector-vector kernel $K^1{\ 1}(x) = \sum_m \Lambda^1_{m1} w^1_{m1} \mathcal{K}(x)^{(m)}$ depends on precisely these grades (with $\Lambda^1_{11} = 0$ by the selection rules of the geometric product), it collapses to $h_0(|x|_\eta)$ — a scalar function of the norm with no directional dependence. This holds for any $\operatorname{O}(p,q)$ just as it does for $\operatorname{O}(2,0)$.
>
> For the conditional kernel network $\hat{\mathcal{K}}$, introducing an auxiliary multivector $\zeta \in \operatorname{Cl}(\mathbb{R}^{p,q})$ with $\langle\zeta\rangle_1 \not\propto x$ yields: grade 0 now contains $\eta(x, \langle\zeta\rangle_1)$, linear in the directional components of $x$, and grade 2 now contains $x \wedge \langle\zeta\rangle_1 \neq 0$. Both channels that were blocked in the standard $\mathcal{K}$ now carry directional information, so $K^1_{\ 1}(x)$ can represent direction-dependent maps.
>
> Furthermore, by stacking $L$ such cross-products across layers of the conditional kernel network, the scalar grade can contain degree-$L$ polynomials in the components of $x$, which is sufficient to represent angular frequency $L$. This is how the frequency-2 components missing in the example $\operatorname{O}(2,0)$  case (Section 3.4) are recovered.
>
> Note that this result specifically addresses the vector-vector interaction ($k = n = 1$). For higher-dimensional algebras, additional grade-to-grade interactions exist. Thus, proving completeness across all of them for general $\operatorname{O}(p,q)$ remains an open part of the conjecture, as we discussed in the Limitations section. However we hope this analysis represents the general case more clearly.
>
> **2. Source of performance gains**:
>
> We appreciate this question. Our ablation study in Appendix D.1 clarifies exactly this distinction. To isolate the impact of recovering the missing kernel bases, we tested conditioning the network on a fixed random vector. Since this random condition contains no information about the input field, any performance improvement must stem purely from the restored degrees of freedom in the kernel. This alone drops the error on SWE-1 from 0.5699 (standard CSCNN) to 0.3225. Switching from the random vector to mean pooling further reduces the error to 0.2982. So while the pooling mechanism does provide usefull global information, the majority of the performance gain is driven by the recovery of the missing steerable kernel components.
>
> **3. a) Parameter-matched SWE-5 comparisons**:
>
> The reason we did not enforce strict parameter matching on SWE-5 is that the baselines for this task were taken directly from (Wang et al., 2024) to ensure a fair and reproducible comparison with published numbers, and these baselines span a wide range of model sizes by design. In the small model regime, C-CSCNN (10M) already outperforms the closest parameter-matched baseline, CViT-S (13M), by a meaningful margin (3.51% vs 4.47%). We therefore expect that under strict parameter matching, C-CSCNN would remain competitive.
>
> **3. b) Why Transolver outperforms C-CSCNN on Maxwell tasks**:
>
> We attribute this to the difference in how both methods capture spatial interactions. Transolver uses global physics attention, which can model arbitrary long-range dependencies between all spatial locations. Electrodynamics are governed by non-local phenomena (e.g., electromagnetic fields propagating across the entire domain), where this global receptive field is particularly advantageous. Our convolutional approach has a local receptive field, and our current conditioning operator (mean pooling) provides only a coarse global summary. As we note in Section 5.3, on fluid dynamics tasks (NS, SWE-1), where dynamics are predominantly local, C-CSCNN outperforms Transolver. We believe that introducing more expressive conditioning operators that capture finer spatial structure (as discussed in Section 6) can bridge this gap on non-local tasks.

---

> > ### Author Rebuttal · Reviewer_oRKz · 2026-04-04
> >
> > Thank you to the authors for the detailed rebuttal. I appreciate the additional clarification on the general $O(p, q)$ case and the other responses. However the new argument still focuses primarily on the vector-vector interaction, while completeness across general grade-to-grade interactions remains open (as also acknowledged in both the rebuttal and the paper's conjectural formulation).  As a result the main theoretical claim that the conditional construction broadly resolves the kernel incompleteness issue remains only partially established.  The rebuttal strengthens the diagnosis and provides a partial remedy, but does not fully close the overall case. Given this, I maintain my score.

---

### Official Review · Reviewer_h6YC · 2026-03-23

**Soundness:** 2
**Presentation:** 1
**Significance:** 1
**Originality:** 3
**Overall Recommendation:** 4
**Confidence:** 3

**Summary:**

The authors aim to discuss the concept of improving equivariant neural architectures for PDE modeling by addressing expressivity limitations in Clifford-Steerable CNNs. The authors examine a general area of geometric deep learning under symmetry constraints, specifically focusing on pseudo-Euclidean equivariance and its impact on learning expressive convolutional operators. The paper identifies kernel incompleteness as a fundamental bottleneck and proposes conditional kernels as a principled solution. The proposed method is validated on diverse PDE modeling benchmarks.

**Compliance With Llm Reviewing Policy:**

Affirmed.

**Final Justification:**

The authors provided a strong rebuttal that addressed most of my initial concerns. In particular, my primary reservations regarding the clarity and presentation of the paper were adequately resolved through their responses. I no longer have objections to accepting this work.

**Key Questions For Authors:**

All questions are addressed in the above weaknesses section.

**Limitations:**

Yes

**Strengths And Weaknesses:**

-Soundness-
* The completeness of the kernel basis is only proven for O(2,0) and remains questionable for general O(p,q).
* The equivariant property is only approximate for discretized grids.

-Presentation-
* Overall, the manuscript is hard to follow and missing details. For example, introduction does not fully demonstrate why models should respect the symmetric nature of physical systems. Motivation behind the need of equivariant network is not fully elaborated.
* The Clifford Algebra is one of the most core parts of the proposed method. However, its background and motivation is not fully addressed throughout the paper. Also, the related works section lack the explanation of Clifford algebra.
* The Examples highlighted in bold under section 3 and 4 looks odd. Better titles can be introduced here.
* The experiments section clearly needs rearrangements and polishing. The author claims they have compared the proposed work with 15 strong baselines, but none of the tables in the main manuscript show 15 comparison models. It is hard to follow which table addresses which capability of the proposed work. Table in general has a poor readability as well.

-Significance-
* The authors studied equivariant nature of geometric deep learning in terms of expressivity. This opens a new perspective towards significance of equivariant nature of neural networks.
* Only simple conditioning such as mean pooling is explored. Any room for improvements through other operations?
* Performance gains of the proposed method is minimal. Transolver work often maintains superior error rates than C-CSCNN for some tasks. Also, in table 2, there are many other works with lower error rates.
* The performance gain from larger C-CSCNN looks minimal compared to smaller C-CSCNN. Thus, the scalability benefits look skeptical. Have C-CSCNN with larger parameter than 55M tested out?

-Originality-
* The explicit analysis of missing basis components is insightful. However, although the missing bases can be recovered through consecutive convolutions, this solution is not explored by the authors and not compared with the proposed method.
* The authors explored bridging linear equivariant convolutions with nonlinear equivariant operators, which is non-trivial.

---

> ### Author Rebuttal · Authors · 2026-03-31
>
> We thank the reviewer for the comments. We address each concern below.
>
> ## Soundness
>
> **1. Completeness for general $\operatorname{O}(p,q)$:** Since the initial submission, we have formalised the incompleteness and its recovery into a general result for the camera-ready version. Due to space constraints, the full derivation is in our responses to Reviewers oRKz and FfRh.
>
> **2. Approximate equivariance on discretised grids:** This is a general consequence of discretising continuous equivariant methods, not specific to our approach. The relative equivariance error of our conditional convolution is $3.4 \times 10^{-7}$, seven orders of magnitude below prediction errors in Table 1, and of the same order as the unconditional CSCNN ($2.4 \times 10^{-7}$, Section 5.3).
>
> ## Presentation
>
> **3. General clarity:** We would appreciate the reviewer specifying which parts are hard to follow so we can address them concretely.
>
> **4. Motivation for equivariance:** We will add to the Introduction: *"By restricting the hypothesis class to functions that respect the symmetries of the problem, equivariant models need not learn these relations from data alone, which has been shown to improve data efficiency and often accelerate convergence [1, 2]."*
>
> **5. Clifford algebra background:** We will (1) add a new Section 3.2 ("Clifford algebra and multivector fields") with the formal definition of the algebra, the geometric product, multivectors, and grades, and (2) add a sentence to the Related Works clarifying why Clifford algebra is useful: *"By encoding scalars, vectors, and higher-order geometric objects as elements of a single algebra, Clifford-based networks can learn interactions between features of different geometric types through the geometric product, enabling them to exploit the structure of the data more effectively [3]."*
>
> **6. Bold "Example" titles:** With the general $\operatorname{O}(p,q)$ result added to the main body, the $\operatorname{O}(2,0)$ case will move to the Appendix as an illustrative example, making this naming appropriate.
>
> **7. 15 baselines:** The 15 refers to the total across all tasks. We will clarify: *"...a total of 15 strong baselines across all tasks."*
>
> **8. Table-to-capability mapping:** We have added an introductory paragraph to Section 5 outlining which tables/figures correspond to which evaluation criteria (Data efficiency → Table 1/Figure 4, Scaling → Table 2, Equivariance → Section 5.3).
>
> **9. Table readability:** We kindly ask the reviewer to clarify which table(s) they are referring to so we can make targeted improvements.
>
> ## Significance
>
> **10. Simple conditioning operators:** The focus of this paper is restoring the missing degrees of freedom and showing this yields significant gains at minimal cost. Simple operators (mean/max pooling, random vector) serve as proof of concept (Appendix D.1). Richer operators such as learnable pooling or localized conditioning (bridging with hierarchical methods like OctFormer [4] and Erwin [5]) are a promising future direction discussed in Section 6.
>
> **11. Performance gains:** We respectfully disagree that the performance gains are minimal. Compared to the original CSCNN, C-CSCNNs achieve drastic improvements across all tasks (Table 1), and outperform all baselines including Transolver on the fluid dynamics tasks (NS, SWE-1). Transolver leads on electrodynamics, which we attribute to its global attention being better suited for non-local phenomena; our simple mean pooling operator captures only global statistics, and more expressive operators (Section 6) can likely close this gap. Regarding Table 2, the baselines with lower error rates are up to 40x larger than our model. At comparable parameter counts, C-CSCNN consistently outperforms alternatives, and even our 55M variant outperforms models like FNO (268M) and UNO (440M).
>
> **12. Scalability:** The modest gain from 10M to 55M reflects saturation of the mean pooling operator, not the architecture. Mean pooling compresses all spatial information into a single vector, limiting what additional parameters can exploit. More expressive operators (Section 6) would allow larger models to leverage their capacity.
>
> ## Originality
>
> **13. Consecutive convolutions as alternative:** Recovering missing bases via consecutive convolutions doubles parameters and memory (Section 1). Our conditioning adds negligible overhead (Section 5.2). Given this cost discrepancy, we did not consider it a viable baseline.
>
> [1] Brehmer et al. (2025). Does equivariance matter at scale? TMLR.
>
>  [2] Weiler, M., Forré, P., Verlinde, E., & Welling, M. (2023). Equivariant and coordinate independent convolutional networks. A Gauge Field Theory of Neural Networks, World Scientific.
>
> [3] Brandstetter et al. (2023). Clifford Neural Layers for PDE Modeling. ICLR.
>
> [4] Wang, P. S. (2023). OctFormer. ACM TOG (SIGGRAPH), 42(4).
>
> [5] Zhdanov et al. (2025). Erwin: A Tree-based Hierarchical Transformer. ICML.

---

> > ### Author Rebuttal · Reviewer_h6YC · 2026-04-04
> >
> > I appreciate the authors for their response. Most of my confusions have been resolved. I will consider updating the score.

---

> > > ### Author Response · Authors · 2026-04-07
> > >
> > > Dear Reviewer h6YC,
> > >
> > > We sincerely thank you for your time and for the thoughtful feedback throughout the review process. We are very pleased that our rebuttal and revisions have fully addressed your concerns and clarified the points raised.
> > >
> > > Thank you again for helping us improve the quality of our work.
> > >
> > > Best regards,
> > > The Authors

---

### Decision · Program_Chairs · 2026-04-30

**Decision:**

Accept (regular)

**Comment:**

All four reviewers support acceptance. They found the theoretical insight convincing and the fix elegant: proving a representational gap exists and closing it with negligible overhead. They found the ablation particularly convincing in showing that most of the gain comes from recovering missing kernel bases, not from the pooling mechanism itself. The incompleteness proof only covers O(2,0) fully, but reviewers agreed this does not undermine the contribution. I recommend acceptance.